



**TITLE PAGE:**
**Dark septate endophytic fungi associated with pioneer grass inhabiting volcanic deposits**
**and their functions in promoting plant growth**
HAN SUN[1,2*], TOMOYASU NISHIZAWA[1,2], HIROYUKI OHTA[1,2], KAZUHIKO NARISAWA[1,2]
[1]*United Graduate School of Agricultural Science, Tokyo University of Agriculture and*
*Technology, 3-5-8 Saiwai-cho, Fuchu-shi, Tokyo 183-8509, Japan*
[2] *Ibaraki University College of Agriculture, 3-21-1 Chuo, Ami-machi, Ibaraki 300-0393, Japan*
**Running Headline:**    Endophytic fungi in volcanic deposits
**Type of the Study:** Original Research Article
***Corresponding Author*:**
Han Sun
E-mail: sunh1211@163.com
Tel: +81-029-888-8667
Fax: +81-029-888-8667





**Dark septate endophytic fungi associated with pioneer grass inhabiting volcanic deposits**
**and their functions in promoting plant growth**
HAN SUN[1,2*], TOMOYASU NISHIZAWA[1,2], HIROYUKI OHTA[1,2], KAZUHIKO NARISAWA[1,2]
[1]*United Graduate School of Agricultural Science, Tokyo University of Agriculture and*
*Technology, 3-5-8 Saiwai-cho, Fuchu-shi, Tokyo 183-8509, Japan*
[2] *Ibaraki University College of Agriculture, 3-21-1 Chuo, Ami-machi, Ibaraki 300-0393, Japan*

**Abstract**
Growth of the pioneer grass *Miscanthus condensatus,* one of the first vegetation to be
established on volcanic deposits, is promoted by root-associated fungi, particularly dark septate
endophytes (DSE). Fungal taxa within DSE colonize the root of *Miscanthus condensatus* in
oligotrophic *Andosol*, and their function in plant growth promotion remains largely unknown.
We, therefore, comprehensively assessed the composition of the DSE community associated
with *Miscanthus condensatus* root in volcanic ecosystems using the approaches of both
metabarcoding (next-generation sequencing) and isolation (culturing). Also, their promotion
effects of DSE on plant growth (rice as a proxy) were evaluated by inoculation of core isolates
to rice roots. Here, we found: i) 70% of culturable fungi that colonized *Miscanthus condensatus*
phylogenetically belonged to DSE, ii) 7 orders were identified by both sequencing and culturing
methods, and iii) inoculation of DSE isolates (*Phialocephala fortinii, P. helvetica, and*
*Phialocephala sp.*) validated their effects on rice growth, particularly under an extremely low
pH condition (compared to control without inoculation, rice biomass enhanced by 7.6 times
after inoculation of *P. fortinii*). This study helps improve our understanding of the community
of *Miscanthus condensatus*-associated DSE fungi and their functions in promoting plant growth.

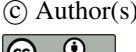

**Key words:** *volcanic deposits*, *pioneer grass*, *dark septate endophytic fungi*, *culture-non-*
*culture approaches*

**Introduction**
Numerous studies demonstrated that symbiotic fungi play a significant role in the
establishment of pioneer vegetation in harsh environments or agricultural soils with extremely
low pH. The association of these fungal micro-organisms that promote plant colonization in
extreme conditions is mainly to: improve host nutrient uptake (Usuki and Narisawa, 2007;
Yadav *et al*., 2009), defend against pathogens (Busby *et al*., 2016), promote tolerance to abiotic
stress (Rodriguez *et al*., 2008; Gill *et al*., 2016), and modify trophic interactions (Clay, 1996;
Omacini *et al*., 2001; Bultman *et al*., 2003).
One of the most common groups of monocotyledonous root endophytes is dark septate
endophytes (DSE), which usually colonize in tissues intracellularly and intercellularly of more
than 600 living herbaceous and woody plant species (Jumpponen and Trappe, 1998). DSE,
which are characterized by their morphology of melanized, septate hyphae and structure like
microsclerotia, also confer the ability to improve plant performance through enhanced nutrient
uptake, and increased ability to withstand adverse environmental conditions (Khastini and
Jannah, 2021). Increasing evidence shows that DSE gradually become the most prevalent root
colonizers under extreme environmental conditions of different ecosystem (Deram *et al*., 2008;
Regvar *et al*., 2010). For example, Huusko *et al*. (2017) reported DSE-dominated colonization
in *Deschampsia flexuosa* roots along a postglacial land uplift gradient. Gonzalez Mateu *et al*.
(2020) reported that DSE inoculation *Phragmites australis* had higher aboveground biomass
under mesohaline conditions. DSE, e.g., *Phialocephala fortinii*, promote host plant growth and



adaptation to the hostile environment by: i) increasing resistance to heavy metal contamination
and heat/drought stress via producing melanized cell walls and, ii) facilitating uptake of
nutrients such as nitrogen and phosphorous (Jumpponen *et al*., 1998; Surono and Narisawa,

2017).

Wild plant species may live in symbiosis with mycoflora that may have been lost during

breeding of the cultivars used in agriculture (Yuan *et al*., 2010). Whilst, some of symbiotic
fungi, that can assist plants to adapt to a given stress in a natural habitat, might increase
tolerance of crop species to that stress in an agriculture system. Thus, from an agricultural point
of view, the plant symbiotic fungi could be seen as an extended source for crop adaptation and
growth in agronomy. In attempts to domesticate "wild" symbiotic fungi, some of these DSE
species in natural system have been successfully transferred to agricultural species from their
original host, providing benefits to the inoculated crops.

Rice (*Oryza sativa*) is the principal food grain crop (one of the four major food crops) for

more than 3 billion people, and its consumption exceeds 100 kg per capita annually in many
Asian countries (Yuan *et al*., 2010). During the last several decades, there have been major
climatic events, including global warming, soil acidification, etc, that decreased agricultural
productivity of rice around the world. Soil pH is a highly sensitive factor to determine plant
survival, distribution, and interactions with microorganisms, which are vital for the availability
of essential nutrients and plant growth (Luo *et al*., 2013). Acidic soils occupy around 40-50%
of the world's potentially arable land. Plants commonly encounter deficient and toxic levels of
mineral elements (soluble ionic Al, mainly $Al^{3+}$)when grown in acidic (pH<5) soil. About 13%
of the world's rice is produced in acid soil. Compared with other crops, rice has relatively
stronger Al toxic resistance (Famoso *et al*., 2010), and is also the most complex cereal crop
with Al resistance genes (Ma *et al*., 2002). Nevertheless, as for other crops, heavy metal toxicity



in acid soil limits rice growth and nutrients uptake, and subsequently reduces grain yield (Chen
*et al.*, 2020). The optimal pH range for rice growth is 5.0-8.5, which shows the likely reduction
of yield in the soil with the extended pH range. To improve these soil acidity, liming is often
used but is practically difficult and unsustainable.
Microorganisms inoculation is a sustainable approach to potentially promote plant
resistance to acidic stress. For instance, plant-associated fungi, such as arbuscular mycorrhizal
fungi (AM fungi), reportedly play a key role in the protection of plants in acidic soils. Yet, high
concentrations of $H^+$ and $Al^{3+}$ can inhibit hyphal growth and spore germination in AM fungi,
thereby decreasing the possibility of colonizing plant roots (Clark, 1997; Van Aarle *et al.*, 2002;
Postma *et al.*, 2007). Comparably, DSE show marked potential to help host plants resist acidity
because of their higher $H^+$ tolerance than other colonizing fungi (Postma *et al.*, 2007). Still,
there is a lack of reports of DSE improving host crop (e.g., rice) growth under acidic conditions,
especially an extremely acidic condition (pH 3.0).
Re-vegetation in volcanic soil, characterized by a dominance of biological processes, is
difficult due to: i) strong acidity of volcanic deposits, ii) high concentration of toxic elements,
and iii) deficiencies in essential nutrients. *Miscanthus sinensis,* an unique pioneer grass plant
during recovery after volcanic eruption, is the first to be established on volcanic deposits, and
frequently found as primary vegetation in lahar deposited by volcanic eruptions (Watanabe *et
al.*, 2006; Hirata *et al.*, 2007; An *et al.*, 2008; Ezaki *et al.*, 2008). This is because *M. sinensis*
can tolerate a wide range of environmental stresses due to the trait of C4 photosynthesis, leading
to high productivity and low-nutrient requirement (Stewart *et al.*, 2009). Apart from *Miscanthus*
traits that adapt to the volcanic soil, the root-associated fungal communities are widely reported
to benefit the growth and promote the adaptation of host plants to stress, such as aridity (Wu
and Xia, 2006), salinity (Porcel *et al.*, 2012), and oligotrophic conditions (Jeewani *et al.*, 2021).



A better understanding of plant-microbe interactions, therefore, can help improve our
understanding of vegetation recovery and plant growth promotion including agricultural
application scene. The isolation and culture of fungal species, therefore, are indispensable as
they complement taxonomic databases and validate taxa revealed by sequencing. Bai *et al*.
(2015) established *Arabidopsis* root-derived bacterial culture collections representing the
majority of species that were reproducibly detectable by culture-independent community
sequencing. Laval *et al*. (2021) investigated fungal and bacterial communities in soils receiving
wheat and oilseed rape residues, and confirmed the feasibility of combined culture-unculture
approaches that revealed consistent community profiles. The role of keystone taxa revealed by
the sequencing data-based co-occurrence network can be further validated by culturing and
followed inoculation. For example, isolation was used to test whether the interaction between
micro-organisms predicted by metagenomic sequencing actually occurs (Laval *et al*., 2021). By
isolation and inoculation, (Zheng *et al*., 2021) identified the strong decomposition ability of
keystone taxa such as the genera *Chryseobacterium* (bacteria), *Fusarium*, *Aspergillus*, and
*Penicillium* (fungi), which are consistent with the keystone taxa revealed by the co-occurrence
network. The combination of sequencing and culturing methods, therefore, is powerful for the
identification of putative taxa (either individually or creation of synthetic communities). Yet,
studies on DSE in volcanic ecosystems by culture-unculture approaches are lacking, and
inoculation to validate the function in rice growth still awaits further investigation.
For this purpose, both culture-dependent and culture-independent approaches were
adopted, to comprehensively reveal the fungal communities of *Miscanthus*-associated,
particularly DSE, from volcanic ecosystems. Their functions in promoting plant growth (via
isolation-inoculation) in different pH soils were further evaluated. Here, we sampled soil and
plants from Miyake-jima, as a model active basaltic volcanic island with an eruption in 2000.



It is located in the Pacific Ocean, and it ejected large amounts of volcanic ash and gases such
as sulfur dioxide and hydrogen sulfide (60% of vegetation on the island was affected). As a
result of $SO_2$ gas exposure, volcanic ash deposits were acidified due to $SO_4^{2-}$ absorption. They
were characterized by strong acidity, with high levels of exchangeable $Ca^{2+}$ and $Al^{3+}$ (Fujimura
*et al.*, 2016). This study, therefore, aimed to: i) reveal the fungal taxa associated with the roots
of *M. condensatus* during vegetation recovery by a combination of sequencing and culturing
approaches, and ii) inoculate the major food crop i.e., rice with these indigenous isolates
(overlapped with sequencing-revealed taxa) to evaluate their effects on rice growth, in
particularly under low pH condition. We hypothesized that abundant colonization by DSE fungi
occurs in the pioneer grass *M. condensatus* inhabiting volcanic deposits near the crater of
Miyake-jima, due to DSE's traits of preferential colonization under oligotrophic and acidic
conditions.

**Materials and Methods**
**Study site description and root sampling**

Miyake-jima (55.5 km$^2$ in area; highest point, 775 m), a basaltic volcanic island (34º05′ N,

139º31′E; Fig. 1), belongs to the Fuji volcanic southern zone in the East Japan volcanic belt.
Mount Oyama, located in the center of the island, is an active volcano. A large amount of
volcanic $SO_2$ gas (~54 kt d$^{-1}$) was ejected immediately from a newly created summit caldera
after the latest eruption in 2000 (Fujimura *et al.*, 2016). The $SO_2$ gas exposure declined slowly
after the eruption, and as a result of this exposure, the volcanic ash deposits were acidified due
to $SO_4^{2-}$ absorption. They were characterized by strong acidity [pH ($H_2O$), 3.1-4.0], with high
levels of exchangeable $Ca^{2+}$ (33.5-115 cmolc kg$^{-1}$) and $Al^{3+}$ (0.8-10.2 cmolc kg$^{-1}$) (Fujimura *et*
*al.*, 2016). At 18 years after the eruption, the patchy vegetation of a pioneer grass, *Miscanthus*



*condensatus*, was established at site OY near the Miyake-jima summit crater (34º04.69′ N,
139º31.04′ E; 553m a.s.l; Fig. 1). The rhizosphere soils of *Miscanthus condensatus* were
collected at site OY in November 2017, and March and September 2018. From each period,
three healthy specimens of *M. condensatus* were collected, kept in sterile plastic bags, and
immediately stored on ice. Samples were divided into two portions, and: 1) kept at 4ºC and
processed within 48 h after collection for isolation, and 2) kept at -20 ºC until DNA extraction
and molecular analysis.

**Root surface sterilization and culturable endophytic fungal isolation**
In order to remove adhering soil and free-living microbes, which are unlikely to interact
with the roots of plants, root samples were gently rinsed with tap water. Individual roots were
severed aseptically in 1-cm-long sections with a sterile scalpel and put into 50-mL conical
centrifuge tubes. Then, they were superficially sterilized with 0.005% Tween 20 and then rinsed
with sterilized distilled water before the aseptic stepwise sterilization process was carried out.
Root sections were treated with 70% ethanol for 1 min, with a further step in the above process,
1% sodium hypochlorite was added and sterilized for 5 min. Finally, sections were rinsed with
sterilized distilled water three times. After surface sterilization, the final wash was spread plated
onto 1/2 Corn Meal agar medium (cornmeal, Difco 25 g $L^{-1}$, Bacto agar, Difco 15 g $L^{-1}$) to
confirm the disinfection and incubated for 2 weeks at 23ºC to examine for the presence of a
growth colony. Root sections were dried with sterile filter paper overnight and then placed onto
cornmeal agar medium containing 0.1 mg $kg^{-1}$ streptomycin and incubated at 23ºC for 2 weeks.
After incubation, pure cultures were obtained by transferring single hyphae to cornmeal malt
yeast agar medium (CMMY; Malt extract 10 g $L^{-1}$, Yeast extract 2 g $L^{-1}$, Cornmeal 8.5 g $L^{-1}$,
Bacto agar 7.5 g $L^{-1}$).




**Identification of fungal isolates and phylogenetic analysis**

Genomic DNA from each fungal isolate was extracted from mycelium using Prepman
Ultra Sample Preparation Reagent Protocol (Applied Biosystems, California, USA). The
universal primer pairs of ITS1 (5'-TCCGTAGGTGAACCTGCGG-3') hybridize at the end of
18S rDNA, and the primer ITS4 (5'-TCCTCCGCTTATTGATATGC-3') that hybridizes at the
beginning of 28S rDNA was used to amplify fungal isolates (Mahmoud and Narisawa, 2013).
PCR amplification was carried out in a 50-μL reaction mixture containing 1 μL fungal genomic
DNA, 2.5 μL of each primer, 5 μL of 10×Ex Taq buffer, 4 μL of dNTP, 0.25 μL of Ex *Taq*
DNA polymerase, and 34.75 μL of sterilized MilliQ water under thermal conditions of 4 min
at 94ºC, 35 cycles of 94ºC for 35 s, 52ºC for 55 s, and 72ºC for 2 min, and a final extension of
72ºC for 10 min using a Takara PCR Thermal Cycler Dice (Takara Bio INC., model TP 600,
Japan). The PCR products were purified and sequenced using an Applied Biosystems 3130*xl*
DNA sequencer. All sequences obtained were compared with similar DNA sequences retrieved
from the Genbank database using the NCBI BLASTN program.

**Illumina MiSeq sequencing for culture-independent identification**

Roots of November samples were added to 10-mL aliquots of sterile distilled water and
macerated with a pestle and mortar for DNA extraction with DNeasy Plant Mini Kit (Qiagen,
Hilden, Germany) following the manufacturer's protocol. DNA was purified using Ultra Clean
DNA Purification Kit (MOBIO, Carlsbad, CA, USA). Then, DNA was eluted in 50 μL of Tris
and EDTA buffer. A NanoDrop spectrophotometer (NanoDrop Technologies, Wilmington, DE,
USA) was used to quantify the DNA concentration. Finally, DNA samples were stored at -80ºC





before molecular analysis. The second nuclear ribosomal internal transcribed spacer (ITS2)
region of the rRNA operon was targeted using the fungal-specific primer pairs ITS3 (5'-
GCATCGATGAAGAACGCAGC-3') and ITS4 (5'-TCCTCCGCTTATTGATATGC-3')
(Chen et al., 2021). PCR amplification was carried out in triplicate with 50-µL reactions
containing 25 µL of Premix Taq (TaKaRa, Shiga, Japan), 23 µL of sterilized MilliQ water, 0.5
µL of both forward and reverse primers (125 pmol), and 1 µL of template DNA. The PCR
program had the following thermocycling conditions: 35 cycles of denaturation at 94ºC for 30
s, annealing at 54ºC for 30 s, 72ºC for 45 s, and a final extension of 72ºC for 10 min. PCR
products were pooled and their relative quantity was estimated by running 5 µL of amplicon
DNA on 1.5% agarose gel, and products were purified with QIA Quick PCR Purification Kit
(Qiagen, Shenzhen, China). The purified mixture was diluted and denatured to obtain an 8 pmol
amplicon library and mixed with an equal volume of 8 pmol PhiX (Illumina) following the
manufacturer's recommendations in the Illumina MiSeq reagent kit preparation guide (Illumina,
San Diego, CA, USA). Finally, 600 µL of the amplicon mixtures were loaded with read 1, read
2, and the index sequencing primers, and paired-end sequencing (each 250 bp) was completed
on a MiSeq platform (Illumina). The sequencing data were processed using the UPARSE
pipeline ([http://drive5.com/usearch/manual/uparse_pipeline.html](http://drive5.com/usearch/manual/uparse_pipeline.html)). The raw sequences were
subjected to quality control. The singleton and chimeric sequences were removed after
dereplication, and the remaining sequences were categorized into operational taxonomic units
(OUT) with 97% similarity and then assigned taxonomy using the UNITE database
([https://unite.ut.ee/](https://unite.ut.ee/)) .

**Inoculation**





The experiment was conducted as a complete randomized factorial design with two factors.
The first factor had four levels: non-inoculation control or inoculation with three dominant
isolates; and the second factor had three levels: pH 3, pH 4, and pH 5. Each treatment consisted
of four replicates with two plants per pot/replicate, thus totaling 48 experimental pots. Fungal
inoculates were prepared by aseptically growing three dominant DSE isolates on Petri dishes
with oatmeal agar medium (10 g L$^{-1}$ oatmeal and 15 g L$^{-1}$ Bacto agar enriched with nutrients: 1
g L$^{-1}$ MgSO$_4$.7H$_2$O, 1.5 g L$^{-1}$ KH$_2$PO$_4$, and 1 g L$^{-1}$ NaNO$_3$). Due to the host non-specific
character of DSE, rice was chosen as a host plant in this study mostly for its important role in
consumed cereal in the world and it is from the same family as *Miscanthus*. Rice seeds were
surface- sterilized by immersion in 70% ethanol for 2 min, and a solution of 1% sodium
hypochlorite for 5 min with agitation. The sterilized seeds were gently rinsed several times with
sterilized distilled water, then dried overnight, and plated onto 1% water agar medium in Petri
dishes for germination at 30ºC. Following pre-germination, 2-day-old seedlings (two seedlings
per plate) were transplanted as growing fungal colonies on the medium at pH 3, pH 4, or pH 5.
For DSE inoculation, two 5-mm plugs excised from the edge of an actively growing colony on
culture medium were inoculated at a 1-cm range close to the rice seedlings. Seedlings
transplanted onto non-inoculated medium were used as controls. The whole set was placed into
sterile plastic culture bottles and incubated for 3 weeks at room temperature with an 18 h:6 h
(L:D) regimen and intensity of 180 μmol m$^{-2}$s$^{-1}$. Assessed plants were harvested and oven-dried
at 40ºC for 72 h. The shoot and root dry weights of treated plants were measured and compared
with the control.
**DSE root colonization observations**





Root colonization by DSE fungal isolates was observed to confirm whether the selected
DSE colonized the inner roots endophytically. Roots were harvested from plants after 3 weeks
of cultivation. Root systems were washed thoroughly under running tap water to remove
adhering agar, then rinsed with distilled water, and used for root staining. The root samples
were cleared with 10% (v/v) potassium hydroxide in a water bath at 80ºC for 20 min.
Subsequently, roots were acidified with 1% hydrochloric acid at room temperature for 5 min,
then stained with 50% acetic acid solution containing 0.005% cotton blue at room temperature
overnight. Root fragments were placed on a slide glass and covered with a cover glass. Fungal
colonization was observed using a light microscope equipped with an Olympus DP25 digital
camera.
**Statistical analyses**
All statistical analyses were performed in the R environment (version: V4.1.2).
Homoscedasticiy was checked using Levene's test and normality using Shapiro-Wilk's test.
The differences of mean dry biomass between the analyzed traits of the seedlings in different
treatments in this study were calculated and analyzed statistically with two-way analysis of
variance (ANOVA) and Tukey's honestly significant difference test at P-values<0.05.
**Results and Discussion**
**The core fungal taxa identified by both culture-dependent and culture-independent**
**methods**
This study compared the culture-dependent isolates with the fungal taxa revealed by
culture-independent methods. Based on 97% sequence similarity, all reads were clustered into
224 OTUs, and the valid sequences were classified into five phyla, including two major





dominant phyla of Ascomycota (71.5%) and Basidiomycota (17.1%), followed by
Mortierellomycota, Mucoromycota, and Calcarisporiellomycota, while the cultivable
endophytic fungi were classified into 2 different phyla of Ascomycota (97.5%) and
Basidiomycota (2.50%). Fifteen and four classes were detected by culture-independent and
culture-dependent approaches, respectively. Specifically, classes Sordariomycetes and
Leotiomycetes (both belonging to phylum Ascomycota) were the major classes in terms of the
number of OTUs. These data were in agreement with a previous study showing that
Leotiomycetes and Sordarimoycetes were the major classes of endophytic fungi associated with
plants (regardless of plant species, associated host tissue) in acidic, oligotrophic ecosystems
and nutrient-limiting boreal and arctic areas (Arnold *et al*., 2007; Yuan *et al*., 2010; Ghimire *et*
*al*., 2011; Luo *et al*., 2014; Knapp *et al*., 2019).
While looking at the lower level, 27 orders were found by Illumina-based sequencing
analysis, and 10 of them had an average abundance over 1%. Among these orders detected by
sequencing, 7 orders were identified via culture-dependent methods as well (Fig. 2).
Significantly higher proportions of Hypocreales (35.6%), Helotiales (21.2%), and Eurotiales
(13.2%) were observed by Illumina-based analysis (Fig. 2). Through culture-dependent
methods, an abundance of Helotiales (70.0%) occupied the whole community, followed by
Eurotiales (15.0%) and Hypocreales (8.75%). In general, the abundant orders of fungal isolates
also showed abundance in the OTU table generated by high-throughput sequencing. The
overlapping of taxa (Hypocreales and Helotiales) identified by both approaches suggests their
significance and dominance in *Miscanthus condensatus*-associated fungal communities.
Similarly, the key fungal and bacterial community in soils amended with wheat and oilseed
residues were identified via culture and non-culture approaches (Laval *et al*., 2021). Several
other studies also confirmed the feasibility to reveal major microbial taxa and showed the





marked potential of adopting the combination of both culture and non-culture approaches to
identify putative taxa (Laval *et al*., 2021; Bai *et al*., 2015; Zheng *et al*., 2021). Undoubtedly a
combination of culture-dependent and culture-independent methods might provide a powerful
strategy to identify and obtain novel endophytes.
The overlapping order Helotiales identified by both culture-dependent methods was
abundant in the *Miscanthus condensatus*-associated fungal community (Fig. 2). The isolates
including *P. fortinii*, *P. helvetica*, and *Phialocephala* sp. belonged to Helotiales species, which
are highly conserved and found to be co-occurring species in the root symbiont communities
based on Sanger sequencing (Walker *et al*., 2011; Bruzone *et al*., 2015). This study also found
these fungi, *i.e., Phialocephala* sp., *P. helvetica*, and *P. fortinii*, in all samples irrespective of
the sampling period (Table 1). Previous studies isolated *P. fortinii* from the root of *Pinus*
*resinosa* (Wang and Wilcox, 1985), *Vaccinium vitis-idaea*, *Betula platyphylla* var. *japonica*,
*Luetkea pectinate* (Addy *et al*., 2000), *Piceas abies*, *Betula pendula* (Menkis *et al*., 2004),
*Rhododendron* sp. (Grünig *et al*., 2008), *Chamaecyparis obtusa*, and *Rubus* sp. (Surono and
Narisawa, 2017). Yet, the phylogeny and ecological effects of *P. fortiniii* on plant quality still
remain largely unknown (Tedersoo *et al*., 2009). For example, *P. fortiniii* itself is genotypically
diverse and composed of at least 21 morphologically indistinguishable but genetically isolated
cryptic species (CSP) (Grünig *et al*., 2008). Up to seven isolates belonging to *P. fortinii* have
been formally described as CSP (Grünig *et al*., 2008). *Phialocephala helvetica* (sub-species of
*P. fortiniii*) associated with the root of *Picea abies* (Stroheker *et al*., 2021) and *Pinus sylvestris*
(Landolt *et al*., 2020), is regarded as one of the most common CSP. Yet, their functions in
promoting plant growth remain largely unknown.

**Colonization of DSE fungal isolates in plant root**





Isolating and characterizing microorganisms could provide insights into their phylogenetic
identification, physiological properties, and metabolic potentials, which will help understand
the formation, persistence, adaptation mechanisms, and ecological functions of microbial
communities (Li *et al.*, 2019). Therefore, these three most promising isolates of *Phialocephala*
sp., *P. fortinii*, and *P. helvetica*, as typical DSE, were further examined regarding their effects
on growth-promoting activity for plants. Based on the inoculation test, all rice seedlings
exhibited healthy growth throughout the experimental period by fungal isolate × agar pH
interaction (Fig. 3).
After harvesting, the roots were stained with 0.05% cotton blue to determine the endophytism
of DSE isolates. Microscopic observation revealed that all DSE isolates successfully colonized
hair roots of rice seedlings. The hair roots were coated with loose wefts of fungal hyphae. This
feature was identical to that previously described for typical DSE, *i.e.,* they are characterized
by microsclerotia, thick, and darkly pigmented septate hyphae in the hair roots. Non-inoculated
plants as a control showed no DSE colonization. The root colonization pattern was similar in
*P. fortinii* and *P. helvetica*, but the degree of fungal colonization of *Phialocephala* sp. was the
lowest compared with those two isolates. The images show the dense networks of hyphae of
DSE inter- and intracellularly colonizing rice roots (Fig. 4). Very few studies, however,
investigated the role and ecological significance of isolated DSE underlying plant growth.

**The role of isolated DSE in rice growth promotion**
As rice is one of the four major food crops for most Asian people, to domesticate these
isolated "wild" DSE can benefit agriculture production. Thus we transferred these DSE isolates
from their original hosts of *Miscanthus condensatus* to agricultural species (rice). DSE is widely
reported to be characterized with non-host specific, but different host (cross family) may have



different responses (in terms of morphology) to inoculated isolate. For example, *P. fortinii* is
frequently reported in roots and formed typical ectomycorrhizae with members of the Pinaceae
plants (Jumpponen *et al.*, 1998). In contrast, for other family plants, *P. fortinii* is often found to
be an endophytic fungi. In addition, *C. chaetospira* was reported able to develop and form spiral
structures resembling ericoid mycorrhizas within the roots of ericaceous plants (Usuki and
Narisawa, 2005). Whilst *C. chaetospira* colonizing in other host family are characterized by
formation of microsleclerotia-aggregations of irregularly lobed hyphae and dark septate hyphae
growing inter- and intracellulary. Considering both plants, used in this study as a host (e.g.,
miscanthus and rice), belong to the same family of grass with similar host responses to DSE
(showing non-host specific trait), we aimed to test effects of these isolated DSE in crop (rice as
a proxy) growth promotion. Here, we found that shoot biomass of rice inoculated with DSE
isolates increased up to 7.6 times, compared with non-inoculated controls (Fig. 5). The greatest
shoot dry weight was recorded in plants treated with *P. fortinii*, followed by *P. helvetica* and
*Phialocephala* sp. Similarly, *P. fortinii* isolates were used to inoculate asparagus plants and
promote plant growth, e.g., shoot biomass increased by up to 53.5% (Surono and Narisawa,
2017). The beneficial effects of *P. fortinii* on enhancing plant yield have been reported
(Jumpponen *et al.*, 1998; Jumpponen and Trappe, 1998).

This improvement in plant growth may be related to the ability of these isolates to use

organic nitrogen sources under nitrogen-deficient conditions. Low nitrogen uptake by plants is
associated with soil acidity. The presence of *P. fortinii* associated with plant tissues
demonstrated its ability to produce a variety of extracellular enzymes that break down complex
forms of organic matter containing nitrogen and phosphorus (Jumpponen *et al.*, 1998). For
example, *Cladophialophora chaetospira* activates soil nitrogen and promotes aboveground
transfer in Chinese cabbage (Usuki and Narisawa, 2007). Therefore, the most abundant DSE





identified by both culture and non-culture approaches, acting as an important mycorrhizal
symbiont via melanized septate hyphae formation that removed resource limitation, might
promote plant growth. A labeled nitrogen study is required to validate this mechanism.
Rice growth was markedly different depending on the combination of DSE isolates and
pH. Differences in dry weight of DSE inoculated rice compared with non-inoculated rice grown
at pH 3.0 (as high as 7.6 fold) were significantly greater than for those DSE inoculated rice
grown at pH 4.0 and 5.0 (as high as 1.6 fold and 1.2 fold, respectively). In particular, the root
dry weight of *P. fortinii*-treated seedlings was the highest at pH 3.0 with respect to that of the
control. Also, we observed that inoculated species of *Phialocephala* effectively promoted plant
growth, particularly under acidic conditions. The enhanced shoot biomass via DSE isolate
inoculation was most marked in acidic environments, *e.g.*, with 7.6, 1.6, and 1.2 times greater
shoot biomass at pH 3, pH 4 and pH 5, respectively. Less promotion of plant growth by
inoculation with *Phialocephala* at pH 5 compared with 4 and 3 agar indicated that these DSE
isolates likely promote plant tolerance to soil acidity.
Many researchers have reported relatively narrow ranges of pH for the presence or activity
of mycorrhizal fungi in soils (Clark, 1997; Postma *et al*., 2007). This is consistent with the
observation that most colonized isolates associated with plants were found in acidic agar.
Similarly, the colonization of investigated plants with DSE significantly decreased with
increasing soil pH (Postma *et al*., 2007). The mechanisms underlying the promotion of plant
growth by DSE fungal have been addressed. DSE fungal might help adaptability of crop to acid
stress, *i.e.*, low soil pH, and subsequent support of plant growth. The relatively high abundance
of DSE supports host survival in stress habitats mainly via high chitin contents and forming
melanized septate hyphae and microsclerotia in plant roots (Likar and Regvar, 2013). Also, it
might increase the concentration of Mg, known to ameliorate Al toxicity, in the roots of
M.sinensis to decrease Al activity (Haruma et al., 2021).

Here, we validated the effects of these DSE isolates on rice growth, particularly under an

extremely low pH condition, e.g., compared to control without inoculation, rice biomass
enhanced by 7.6 times after inoculation of *P. fortinii*. DSE show great potential to help host
crop resist acidity and thus enable crop cultivation, especially in acidic soil (Postma *et al*., 2007).
Acidic soils occupy up to 50% of the arable worldwide, and around 13% of paddy is acid soil.
While soil acidification can be a problem for crop yield, these DSE isolates might be used as a
management strategy to reduce acidic harm to crops. This, yet, awaits field investigation.

Taken together, this study helps improve our understanding of the community of

*Miscanthus condensatus*-associated DSE fungi and their functions. Our findings suggest that
DSE have the ability to support rice growth under an extremely acidic conditions, and the
formation of melanized septate hyphae and microsclerotia-associated rice tissues might
promote increases in rice growth and root biomass via removing stress and resource limitations,
and thus they show marked potential in not only re-vegetation of pioneer plants in post-volcanic
ecosystems but also promotion of rice growth.

**Conclusion**

The present study provided detailed insights into the diversity and function of the

endophytic fungal community in *Miscanthus condensates*, using both culture-dependent and -
independent approaches. Here, we showed that the fungal community was dominated by
isolates of *Phialocephala*, which were abundant and widely distributed in the volcanic deposits.
Additionally, we validated the functions of these DSE in rice growth, particularly under acidic
conditions, by adopting the approach of isolation-inoculation. Considering that these fungal



isolates promote plant adaptation to acidic soil, the identified DSE, e.g., *Phialocephala. fortinii,*
*P. helvetica*, *and Phialocephala* sp., might be potential candidates as plant growth-promoting
fungi for either restoring vegetation or promoting rice growth under extreme conditions.

**Acknowlegements**

This research was supported by a Grant-in-Aid for Scientific Research (B) (No.21H02191)

from the Japan Society for the Promotion of Science (JSPS), and a Grant-in-Aid for Challenging
Exploratory Research (No.22K19164) from JSPS.

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






**Tables**

**Table 1.** Summary of the endophytic fungal isolates among three months of sampling in *Miscanthus condensatus*

| Phylum | Class | Blast top-hit | Sequence similarity (%) | Accession number in NCBI | Total number | | |
|---|---|---|---|---|---|---|---|
| | | | | | Nov | Mar | Sep |
| *Ascomycota* | *Sordariomycetes* | *Acremonium sp.* | 98 | KT192555.1 | 4 | 2 | 0 |
| | | *Sarocladium sp.* | 99 | MG649463.1 | 3 | 2 | 0 |
| | | *Xylariaceae sp.* | 97 | AB741591.1 | 1 | 1 | 0 |
| | | Arthrinium phaeospermum | 99 | MH857420.1 | 0 | 2 | 2 |
| | *Leotiomycetes* | *Phialocephala fortinii* | 97 | KJ817297.1 | 24 | 17 | 16 |
| | | *Phialocephala helvetica* | 97 | MT107593.1 | 21 | 36 | 37 |
| | | *Phialocephala sp.* | 99 | KT323172.1 | 11 | 14 | 16 |
| | | *Pezicula ericae* | 99 | NR155653.1 | 0 | 5 | 2 |
| | *Eurotiomycetes* | *Talaromyces verruculosus* | 97 | MG748649.1 | 9 | 2 | 2 |
| | | *Penicillium funiculosum* | 97 | JQ724527.1 | 3 | 0 | 0 |
| | *Dothideomycetes* | *Pyrenochaetopsis setosissima* | 97 | LT623227.1 | 2 | 2 | 1 |
| *Basidiomycota* | *Agaricomycetes* | *Tulasnella calospora* | 98 | JQ713577.1 | 1 | 0 | 0 |
| | | *Hypochnicium cremicolor* | 97 | KP814161.1 | 1 | 0 | 0 |
| | | *Phaeophlebiopsis peniophoroides* | 98 | KP135417.1 | 0 | 0 | 3 |
| | | *Phlebiopsis gigantea* | 98 | MH114867.1 | 0 | 0 | 3 |
| *Dikarya* | *Polyporus* | *Polyporus arcularius* | 99 | KP283489.1 | 0 | 1 | 0 |
| *Mucoromycota* | *Mortierellomycotina* | *Mortierellales sp.* | 97 | JQ272348.1 | 0 | 2 | 1 |
| | | | | | 80 | 86 | 83 |


























**Figure legends**

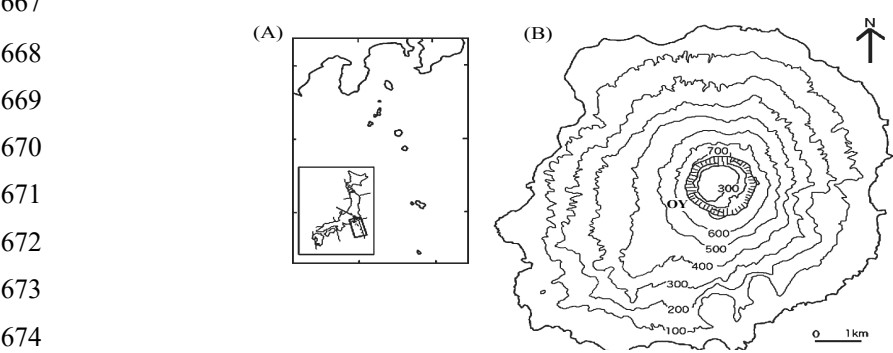


**Fig. 1.** (A) Map showing the location of Miyake-jima in the western rim of the Pacific Ocean. (B) Map showing study site OY
near the summit crater in Miyake-jima

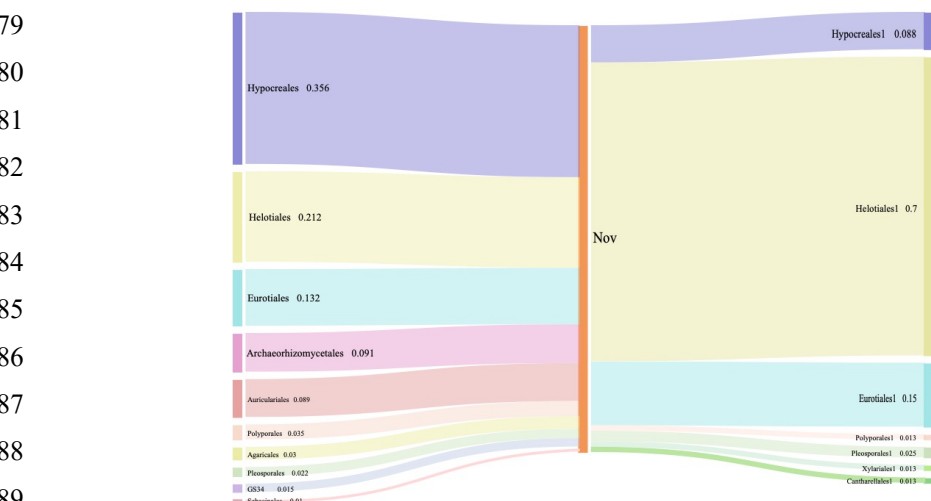


**Fig. 2.** Composition and relative abundance of endophytic fungi at order level by culture-independent (left) and culture-
dependent methods (right)



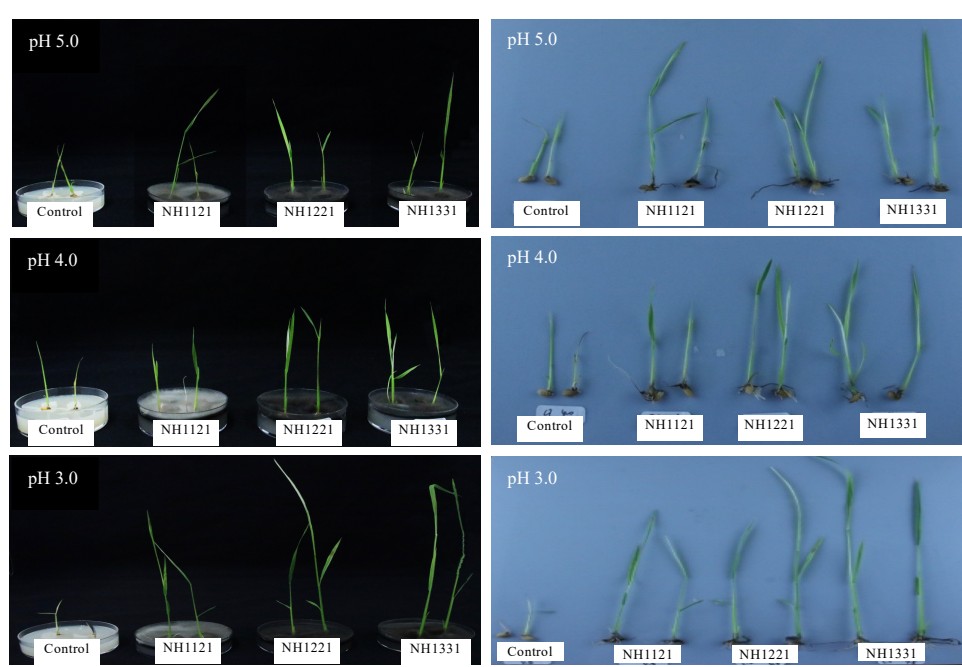

**Fig. 3.** Growth and development of rice plants inoculated with DSE fungal isolates under different pH conditions.

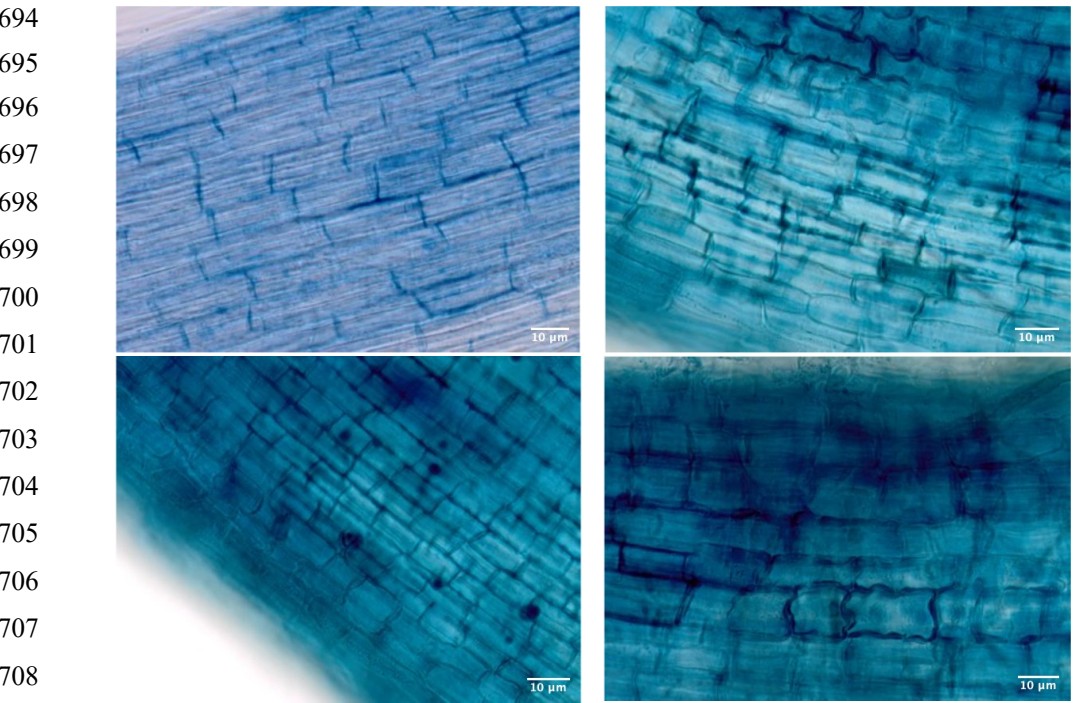



**Fig. 4.** (A) Non-treated DSE as control roots. (B) *Phialocephala* sp. (NH1121)-treated roots. (C) *Phialocephala helvetica*
(NH1221)-treated roots. (D) *Phialocephala fortinii* (NH1331)-treated roots.

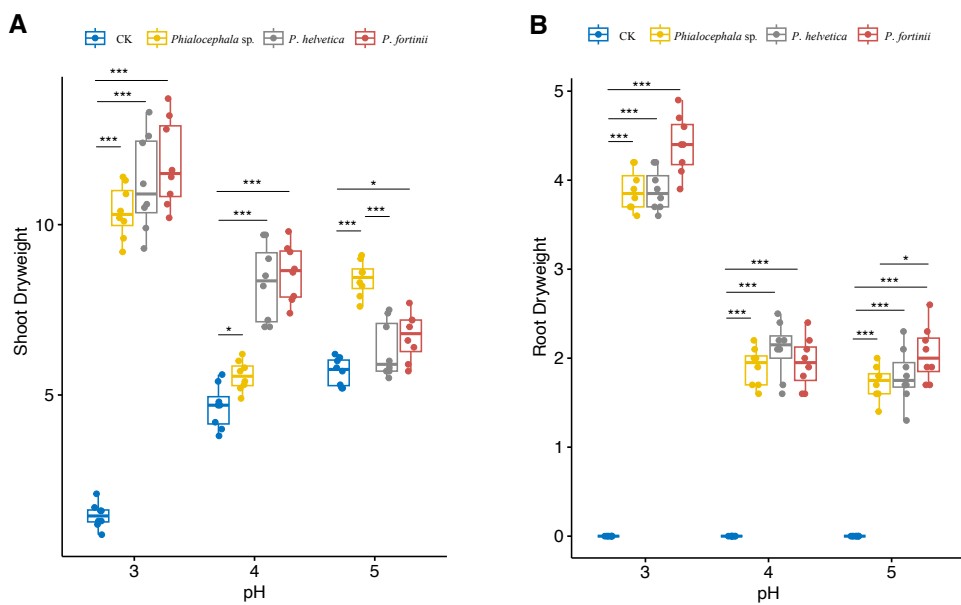


**Fig. 5.** Shoot and root dry weights of rice seedlings inoculated with NH1121 (*Phialocephala* sp.), NH1221 (*Phialocephala*
*helvetica*), and NH1331 (*Phialocephala fortinii*) after three weeks of growth on oatmeal agar either at pH 3, pH 4, or pH 5
(acidic conditions). There are biological replicates (n=8). Median values are lines across the box with lower and upper boxes
indicating the 25th to 75th percentiles, respectively. Whiskers represent the maximum and minimum values. Significance was
determined by ANOVA.