# Peer review of "TITLE PAGE: Dark septate endophytic fungi associated with pioneer grass inhabiting volcanic deposits and their functions in promoting plant growth HAN SUN1,2\*, TOMOYASU NISHIZAWA1,2, HIROYUKI OHTA1,2, KAZUHIKO NARISAWA1,2 1United Graduate School of Agricultural Science, Tokyo University of Agriculture and Technology, 3-5-8 Saiwai-cho, Fuchu-shi, Tokyo 183-8509, Japan 2 Ibaraki University College "

_Biogeosciences, 2023_

## Author Comment (AC1)

**RC1:**

Sun et al., assessed the diversity and function of the DSE community associated with Miscanthus condensatus root in volcanic ecosystems. Volcano on the Miyake-jima island was firstly erupted in 2000. Approaches of metabarcoding (next-generation sequencing) and isolation (culturing) were combined in this paper. This is truly remarkable, given that many recent studies are heavily reliant on sequencing. One of the interesting findings is that 70% of culturable fungi that colonized Miscanthus condensatus belonged to DSE by both sequencing and culturing methods. Also, valuable work is done to validate these DSE isolates (such as Phialocephala fortinii) effects on rice growth by inoculation. These data improve our understanding of Miscanthus condensatus-associated DSE fungi and their functions in promoting plant growth in extreme environments. This paper ought to gain a lot of attention. I recommend the paper for publication with some minor corrections.

Re: Thanks for your positive comments.

The introduction needs to be further improved to be more concise and avoid competition with discussion. And pay attention to the references.

Re: We have removed the competition part between introduction and discussion. Also added the references as suggested below.

Keywords may include: inoculation, plant growth promotion

Re: These keywords are added, thank you.

Line 54, why "promote plant colonization in extreme" is important? I recommend to add with more details.

Re: We have amended it as "The association of these fungal micro-organisms that promote plant colonization is significant in extreme conditions. As these fungal symbionts help plant survival mainly by: improving host nutrient uptake (Usuki and Narisawa, 2007; Yadav *et al*., 2009), defending against pathogens (Busby *et al*., 2016), promoting tolerance to abiotic stress (Rodriguez *et al*., 2008; Gill *et al*., 2016), and modifying trophic interactions (Clay, 1996; Omacini *et al*., 2001; Bultman *et al*., 2003). "

Line 80, what do you mean "wild"?

Re: it means genetically wild type. We have also amended in the text.

Line 81-82: reference

Re:  The reference has been added, thank you.

Line 86: 'decreased' to 'influenced' as the actual yield is increased globally.

Re: It has been revised, thank you.

Line 89-90: reference

Re:  We have removed this sentence to avoid any confusion.

Line 97, why lime is important? this part must be carefully checked and make it more relevant to this study. Also, add a concluding sentence here.

Re: lime has its shortage while it helps alleviate acidic harm. So that is why we suggest fungal inoculation in plant growth in acid soil.

Line 98: '5.0-8.5' is the growth range or the optimal range? Reference

Re:  It is the optimal pH range for rice growth. Reference is added as well. Thank you

Line 101: reference

Re:  It has been added, thank you.

Line 131: change reference format

Re:  We have amended it accordingly.

Line 143-147: move to MM section

Re: We have some introduction in MM section. Here, we would like to keep it as the readers could have a quick understanding of our purpose of this study. Thanks.

I also found this content a little boring. Therefore, follow the referee's instructions and move them to the M & M section.

Line 270, a more relevant subtitle should be provided in the discussion part, some of sentence are introduced in the early part, please avoid repetitive.

Re: I think the subtitle was given in our last version. The repetitive is avoid, as suggested above.

Line 393, please rephrase this sentence.

Re: It has been done, accordingly. Thank you very much.

Fig 1B: Which type of the map? DEM or DSM? Specify it in the caption.

Re: It has been done, thank you.

---

## Author Comment (AC2)

**RC2:**
**Major comments**

This is an interesting research paper of two compared microbial approaches for characterizing some types of endophytic fungi with pioneer grass Miscanthus condensatus. The data indicate that seven orders of DSE were found in the culture method and some isolates have the ability to improve rice growth, especially in extremely low pH condition. Sun and colleague also conduct pot experiments to verify their hypotheses. It's an interesting study, but there are a few issues, which I detail below. The paper could be accepted before the authors make some revisions.

Re: thanks for your relatively positive comments.

First, this paper still needs more supportive references added to the material and methods. When authors do the root surface sterilization, they use 70% ethanol for 1min, which will kill most of the microbes including microbes that are good for plant growth and bad for plant growth. After several sterilization steps, the pure cultures just begin. There are some standard ways or currently- approved approaches to isolate rhizosphere microbiome (like Joseph et al., 2015). I strongly recommend authors give more references to support that the method they have chosen is scientific or recognized.

Re: thank you, we have added some references in this section to validate it is scientifically correct.

Second, the authors have conducted the MiSeq sequencing for identification. It's great. However, I have not found enough analyses about the sequencing data. Why not put the data together into the whole paper? I believe the results and discussion sections could be more abundant.

Re: thank you. This is study combined both sequencing and isolation methods. Also, the inoculation is included. To put many figures in the main text would dilute the whole results. Anyhow, the suggestion is valuable. Thank you.

**Minor comments**

L48 *Miscanthus condensatus* is the subject this research paper focuses on. So add the species name as key words.

Re: I have added, thank you.

L62 which is characterized by

Re:It has been revised, accordingly.  Thank you.

L66 These two references only represent the ecosystem that heavy metal pollution influenced. Add more references to show other different ecosystems.

Re:I have add another reference about the effects of DSE on acidic mine soil.

L72 add more references for i) part.

Re: I have added, thanks.

L143 simplify the wording, here are two "it".

Re:It has been revised, accordingly. Thank you.

L144 how to know 60% of vegetation was affected, add more references to prove or be more specific to describe how to measure this metric.

Re:I have add two references, and they measured the damage rate of vegetation by satellite images (SPOT-2/HRV-XS and TERRA/ASTER).

L151 maybe abundant is not an appropriate word.

Re:It has been revised, accordingly. Thank you.

L160 it's not a clear expression, remove the parenthesis, like "Mount Oyama is an xxx, located in the center of xxx".

Re:It has been revised, accordingly. Thank you.

L165 please check the units. $Ca^{2+}$ cmol kg $^{-1}$, notcmolc $kg^{-1}$?

Re: It was written as cmolc $kg^{-1}$ in the reference, maybe spelling errors, I have revised, thanks.

L167 remove the number after the decimal point.

Re:It has been revised, accordingly. Thank you.

L170-171 how divide three healthy specimens into two portions? it is not reasonable.

Re:I didn't express clearly. Three healthy specimens were collected, and then individual roots of each specimen were divided into two portions for different experimental purpose.

L177 remove the sentence "which are xxx with the roots of plants".

Re:It has been revised, accordingly. Thank you.

L184 upper case Cornmeal.

Re:It has been revised, accordingly. Thank you.

L175-L190 the whole part lacks supportive references.

Re: Thank you for your kind comments. Isolation was performed by modifying the method of Sahu, et al., (2022). Detailedly, the requirement of surface sterilization protocol is different according to different host plant. The sterilant solution, concentration and exposure time could be optimized based on host plant and tissue type. In the present study, the sterilant solution, concentration and exposure time were optimized by a series of gradient experiments (single factor and multilevel design). Based on these pre-experiment (data not shown in this paper), we successful realized the isolation of endophytic fungi (with high diversity but no surface contamination) for this study and future use.

L181 I am not sure 70% ethanol is good enough.

Re: Ethanol acts as lipid solvent and protein denaturant, it is commonly used at con concentration of 60%-85% as a surface sterilant. However, by and large, in the ethanol-sodium hypochlorite combination, the concentration and exposure of ethanol are constant. Around 70%-75% ethanol is used for 0.5-1 min. Since NaOCl is a strong oxidative agent, a lower concentration with short exposure is generally given.

L187 why two weeks incubation? Add more description about the reasons.

Re: The plates were checked every second day for fungal growth. Endophytic fungi were considered have a relatively slow growth rate (about 10-15 mm of radial growth in 7 days). DSE were the most prevalent with the greatest number of isolates. Other taxa were isolated in much smaller numbers. *Mortierella*, *Trichoderma*, *Penicillium*, etc, were mostly isolated within 5-7 days of placing roots in the medium. Generally, 10 days are required to establish DSE colonization of the host plant (Actually, 2-3 weeks has often been used in many other studies for short-term growth evaluation of host plants using DSE). So in this study we incubated 2 weeks for establishing DSE colonization of the host plant.

L188-L190 add more detailed information about incubation.

Re: I have added, thank you.

L208 "November samples" is not an appropriate phrase. It's confusing.

Re: I have revised, thank you.

L227-L229 shorten this sentence. It's tedious.

Re: It has been revised, accordingly. Thank you.

L230 & L234 add reference after website.

Re: This is used very often, we guess the website is also kind of reference. We could add it if reviewer insist.

L238 which three dominant isolates?

Re: I have added, thank you.

L240 remove replicate.

Re: Sorry not clear here, i guess replicate is OK here.

L240 thus totaling is not an appropriate wording.

Re:  sorry, we have corrected to "total" and changed the sentence as "and thus total 48 experimental pots in the study".

L246 suface- what? One more space.

Re:  It has been revised, accordingly. Thank you.

L275 Results and Discussions

Re: It has been revised, accordingly. Thank you.

L283 change 2 to two. Please go over the whole paper like this mistake (like L297 change 7 to seven).

Re: It has been revised, accordingly. Thank you.

L337 sentence-initial needs some space.

Re: It has been revised, accordingly. Thank you.

L345 inter- and intra-celluarly.

Re: It has been revised, accordingly. Thank you.

L414 they? Re-write this sentence.

Re:  Sorry, it is lengthy, we have separated it. Thank you.

**References:**

L459 L462 L464 L466 L470 and others, please uniform the journal name refer to the journal's requirements.

Re: It has been revised, accordingly. Thank you.

L453 L531 L532 species name should be italic.

Re: It has been revised, accordingly. Thank you.

L489 no pages?

Re: It has been revised, accordingly. Thank you.

L542 journal name?

Re: Plant Soil. It is short for the name of journal Plant and Soil.

L548 350 C?

Re: It has been revised to 350 ºC. Thank you.

L549 Soil Biology? And no doi.org is listed.

Re: It has been revised to **Soil Biol Biochem 57, 513-523. DOI: 10.1016/j.soilbio.2012.10.033. Thank you**

L606 Al?Fe ?

Re: Sorry, it has been revised to make it clear. Thank you.

**Figures:**

L691 legend is needed for the figure.

Re: It has been revised, accordingly. Thank you.

L693 I don't know which one picture represents which pH conditions. Add the pH label in the figure and list the pH values.

Re: We have added the pH label in the figure and list the pH values.

L709 please add "*" and "***" means what?

Re: we have added the details. Thank you.